# Trajectory Optimization in a Cooperative Aerial Reconnaissance Model

**DOI:** 10.3390/s19122823

**Published:** 2019-06-24

**Authors:** Petr Stodola, Jan Drozd, Jan Nohel, Jan Hodický, Dalibor Procházka

**Affiliations:** 1Department of Intelligence Support, University of Defence, 662 10 Brno, Czech Republic; jan.nohel@unob.cz; 2Department of Tactics, University of Defence, 662 10 Brno, Czech Republic; jan.drozd@unob.cz; 3Department of Aircraft Technology, University of Defence, 662 10 Brno, Czech Republic; jan.hodicky@unob.cz; 4Centre for Security and Military Strategic Studies, University of Defence, 662 10 Brno, Czech Republic; dalibor.prochazka@unob.cz

**Keywords:** cooperative aerial reconnaissance, unmanned aerial vehicles, simulated annealing, optimization of waypoints, trajectory optimization, experiments, simulation

## Abstract

In recent years, the use of modern technology in military operations has become standard practice. Unmanned systems play an important role in operations such as reconnaissance and surveillance. This article examines a model for planning aerial reconnaissance using a fleet of mutually cooperating unmanned aerial vehicles to increase the effectiveness of the task. The model deploys a number of waypoints such that, when every waypoint is visited by any vehicle in the fleet, the area of interest is fully explored. The deployment of waypoints must meet the conditions arising from the technical parameters of the sensory systems used and tactical requirements of the task at hand. This paper proposes an improvement of the model by optimizing the number and position of waypoints deployed in the area of interest, the effect of which is to improve the trajectories of individual unmanned systems, and thus increase the efficiency of the operation. To achieve this optimization, a modified simulated annealing algorithm is proposed. The improvement of the model is verified by several experiments. Two sets of benchmark problems were designed: (a) benchmark problems for verifying the proposed algorithm for optimizing waypoints, and (b) benchmark problems based on typical reconnaissance scenarios in the real environment to prove the increased effectiveness of the reconnaissance operation. Moreover, an experiment in the SteelBeast simulation system was also conducted.

## 1. Introduction

The use of unmanned aerial vehicles (UAVs) for monitoring, surveillance, and/or reconnaissance operations both in the military and civilian domains has been a rising trend in recent years. Although this trend is relatively new, some of its features are already clear: the technology behind these systems is becoming increasingly mature; the effectiveness of these operations can be rapidly increased by using multiple UAVs that cooperate with one another; and route planning for individual systems is critical in these applications, as it determines the quality of operational performance.

In the military, reconnaissance and surveillance operations serve to increase situation awareness on the battlefield. For planning and executing these operations, Command, Control, Communication, Computer, Intelligence, Surveillance, and Reconnaissance (C4ISR) systems are used. Moreover, these systems often contain decision support features that support a commander in their decision-making processes.

### 1.1. Motivation

At the University of Defence, Czech Republic, the Tactical Decision Support System (TDSS) is being developed. The objective of this system is to support tactical level commanders of the Czech Army in their decision-making process [1]. A number of models of military tactics are included in the TDSS. If one of the models is compatible with the task of the commander, they can use this system for planning; TDSS provides possible variants and options to fulfil the task, including the second-order effects. Thus, the task can be conducted effectively in the optimal or near-optimal manner (according to the selected criterion). More information about this topic can be found in the literature [1,2,3,4,5,6,7,8].

TDSS contains several models used for planning reconnaissance or surveillance tasks via both ground and aerial unmanned systems; for example, cooperative aerial reconnaissance, cooperative ground reconnaissance, cooperative aerial surveillance, and surveillance via a number of unmanned ground sensors (UGS).

Reconnaissance of the area of interest is one of the most frequent tasks of a tactical level commander. Effective reconnaissance can be critical; planning fast and complete aerial reconnaissance using a fleet of UAVs can significantly contribute to the success of the operation at hand. The original aerial reconnaissance model implemented in TDSS has been extended by findings and results from this article, thus increasing its efficiency.

### 1.2. Contribution and Organization of the Article

Cooperative Aerial Reconnaissance (CAR) is the model proposed by the authors in the past. This original model transforms a reconnaissance operation into a problem of planning the trajectories of available UAVs in a graph created from waypoints deployed in the area of interest to be explored. Section 3 revises the basic approach and principles of the CAR model published within the previous research of the authors. Also, the original algorithm used for the waypoints’ deployment appears in this section in order to compare the old approach with the new one.

The contribution of this article is in the substantial improvement of the CAR model. The main idea behind this improvement consists of reducing the number of waypoints that need to be deployed, with the quality worsening or the requirements of the whole operation being undermining without this reduction. This is done by finding the minimal number of waypoints necessary and optimizing their positions. Section 4 is the key part of this article; it introduces the novel approach, formulates the optimization problem, and presents the proposed algorithms for (a) optimizing the number of waypoints and (b) optimizing the positions of waypoints. Section 5 sets out the experiments conducted in order to verify the proposed algorithms.

## 2. Literature Review

Issues around aerial reconnaissance have been discussed in many scientific publications. Paucar et al. [9] deal with the use of UAVs for surveillance and reconnaissance operations in military areas and analyse the benefits of their model in the Ecuadorian Armed Forces. Chen et al. [10] consider a reconnaissance task assignment problem for multiple UAVs with different sensor capacities; the authors propose a modified multi-objective symbiotic organisms search (MOSOS) algorithm to solve the problem. Wang et al. [11] model UAV reconnaissance mission planning as an uncertain multi-objective orienteering problem; to solve this problem, the authors propose a discreet multi-objective bat algorithm with a local search strategy.

Many publications exist that examine route planning for UAVs in various applications. Vivaldini et al. [12] propose a framework for efficient visual data acquisition using UAVs; this combines perception, environment representation, and route planning in the task of disease classification. Liu et al. [13] present a UAV route planning problem for aerial photography under interval uncertainties. The authors of [14] consider the offline route planning of UAVs for a coverage search mission in a river region. The military topic of battlefield situation awareness is covered in the work of [15]; here, the authors propose a hybrid algorithm combining A* and ant colony optimization algorithms for real-time path planning.

Similarly, many publications deal with methods for UAV route optimization. A great overview of methods for trajectory optimization is given in the work of [16]; the methods considered in this paper include linear programming, dynamic programming, genetic algorithms, and neural networks. Different strategies for patrolling tasks conducted by multiple agents on a network defined as a graph are presented in the work of [17]. Turker et al. [18], meanwhile, use simulated annealing for route planning of multi-UAV systems on a graph.

The authors of [19] present a persistent intelligence, surveillance, and reconnaissance routing problem, which includes collecting data from a set of specified task locations and delivering those data to a control station. Zhen et al. [20] consider a vehicle routing problem in which UAVs monitor a set of areas with different accuracy requirements. The problem of how to deploy multiple UAVs most efficiently was considered in the work of [21], in an application where UAVs act as wireless base stations that provide coverage for ground users.

Popular methods often used in applications of reconnaissance, surveillance, or search for an object are based on coverage path planning. An extensive survey of such methods is provided by Cabreira et al. [22]; their article aims at analyzing the existing literature related to the different approaches, including both simple geometric flight patterns and more complex grid-based solutions considering full and partial information about the area of interest. The problem of searching for a single lost target by a single autonomous airborne sensor system using the Bayesian approach was examined in the work of [23]; a stationary or a drifting object at sea was considered as a target. The similar problem of locating a mobile, non-adversarial target in an indoor environment using multiple robotic platforms was investigated by Hollinger et al. [24]; they refer to the problem as multi-robot efficient search path planning (MESPP).

Many publications focusing on monitoring and surveillance using UAVs assume that the ground area to be monitored is perfectly flat. This assumption is not realistic; uneven terrain or high objects (buildings, obstacles) may occlude same parts in the area to be explored. The analysis of this issue is examined by Geng et al. [25] on a problem of continuous surveillance by a group of UAVs in an urban area. The similar problem is considered by Jakob et al. [26]; they present their own approach to plan the aerial surveillance in urban areas via a fleet of fixed-wing UAVs. They propose and compare three planning methods that take into account sensor occlusions caused by high buildings and other obstacles in the target area. Savkin and Huang [27] present their approach to estimate the minimal number of UAVs needed to monitor very uneven target terrain.

Although UAV technology is a very topical issue and thus there are many scientific publications considering various problems from different perspectives, the authors of this article are not aware of any research that deals with the trajectory optimization of UAVs by means of waypoint deployment.

## 3. Cooperative Aerial Reconnaissance

The CAR model is designed for exploring an area of interest by a fleet of unmanned aerial systems (UASs). The objective of the model is to plan the routes of individual UASs so that the area of interest is explored in an optimal or near-optimal manner. The optimization criterion depends on the commander and their task at hand; in most cases, the requirement is to explore the area as fast as possible. The CAR model assumes that the target area is perfectly flat.

The model was mathematically formulated in the work of [28] in detail. The principle is to cover the area of interest evenly, using a number of waypoints so that no space is unexplored when each waypoint is visited during a reconnaissance operation by any UAS available in the fleet. Each waypoint needs to be visited just once. The CAR model plans the trajectories of all UASs according to the selected optimization criterion.

### 3.1. Example Situation

Figure 1 shows an example situation. There are 2 UASs (labelled A and B) available in the fleet deployed in the area of operations. The area of interest to be explored is covered by 27 waypoints (see the green dots in Figure 1a). In Figure 1b, the trajectories (blue lines) of individual UASs are shown, which represent the solution found by the CAR model; as can be seen, the waypoints are evenly distributed between available UASs.

The phase of the CAR model that plans the trajectories for UASs is similar to the well-known multi-depot vehicle routing problem (MDVRP). The difference consists in formulating the optimization criterion. Whereas the objective in the MDVRP is to minimize the sum of lengths of all routes, the objective in the CAR model is alterable according to the requirements of the commander; in most cases, it is to minimize the time needed to complete the longest route by the corresponding UAS.

The original CAR model was improved in the work of [29], where the trajectories were smoothed in order to achieve the following: (a) shorten their length and thus hasten the reconnaissance operation, and (b) be able to use fixed-wing UASs that cannot change direction so abruptly. The result of this improvement, applied to the previous example situation, is shown in Figure 2 in red.

### 3.2. Requirements for the Deployment of Waypoints

The area of interest has to be covered by a number of waypoints so that, after reconnaissance, there is no space in the area left unexplored. This depends on the technical parameters of the sensory systems of UASs, as well as the tactical requirements of the commander. The former concerns the vertical and horizontal field of view and resolution of the camera, while the latter relates to the minimum and maximum permitted height of flight above ground level of UASs in the fleet.

The situation is presented in Figure 3. A UAS flies at the height $H_{AGL}$ above ground level, which is the minimum height allowed. At this height, the camera of the UAS scans the green area (rectangle), the size of which is determined by the angular vertical and horizontal field of view ($A_{VFOV}$, $A_{HFOV}$). The resolution and other technical parameters of the camera must be adequate to be able to scan the area at the required level of detail from the maximum allowed height of flight.

The width and length of the scanned rectangle, $D_{W}$ and $D_{L}$, respectively, are calculated according to Formula (1).

(1)$$D_{W}=2\cdot H_{AGL}\cdot \tan\left(\frac{A_{HFOV}}{2}\right),\;D_{L}=2\cdot H_{AGL}\cdot \tan\left(\frac{A_{VFOV}}{2}\right),$$
where $D_{W}$, $D_{L}$ are the width and length, respectively, of the scanned area; $H_{AGL}$ is the minimum flight height above the ground level; and $A_{HFOV}$, $A_{VFOV}$ are the angular horizontal and vertical fields of view, respectively.

To ensure that no space is left unexplored after the reconnaissance operation, the maximum permitted distance $D_{max}$ from whichever point lying inside the area of interest to its nearest waypoint must be calculated according to Formula (2). When calculating $D_{max}$, the smaller value of the width and length $D_{W}$ and $D_{L}$ is used. This is because the trajectories of the UASs (i.e., their pitch when located at individual waypoints) are not known at the time when the waypoints are deployed; therefore, the smaller (i.e., worse) value must be used.

(2)$$D_{max}=\frac{\min\left(D_{W},D_{L}\right)}{2},$$
where $D_{max}$ is the maximum permitted distance from whichever point inside the area of interest to the nearest waypoint, and $D_{W}$, $D_{L}$ are the width and length, respectively, of the scanned area.

### 3.3. Waypoint Deployment in the Original CAR Model

A simple algorithm was used in the original CAR model to deploy the waypoints. This algorithm is presented in Figure 4. The waypoints are inserted along individual edges of the polygon, forming the area of interest.

The algorithm is fast and straightforward. However, it does not guarantee that the maximum distance requirement is ensured from all points in the area of interest. The result can be influenced by playing with distance $d_{w}$ (see point 2 in Figure 4) and repeating the whole algorithm. It was empirically found that the $D_{max}$ requirement is met in most cases when $d_{w}=0.8\cdot D_{max}$.

## 4. Trajectory Optimization

This section deals with improving the CAR model by optimizing the number and positions of waypoints in the area of interest.

### 4.1. Optimization Problem

The optimization problem consists of finding the minimum possible number of waypoints, provided that condition (3) is not violated.

(3)$$d_{max}\leq D_{max},$$
where $d_{max}$ is the longest existing distance from a point in the area of interest to the nearest waypoint, and $D_{max}$ is the maximum permitted distance introduced in Section 3.2 (see Formula (2)).

The algorithm calculating the minimum number of waypoints n is shown in Figure 5. It is based on the principle of the half-interval search. Firstly, the limits $n_{min}$ and $n_{max}$ are determined; the lower limit $n_{min}$ is set to 1 and the higher limit $n_{max}$ is calculated according to Formula (4), which ensures that condition (3) is met with certainty. Then, the limits gradually approach one another until they meet, which means that the minimum possible number of waypoints is found.

(4)$$n_{max}=⎡\frac{w}{\sqrt{3}\cdot D_{max}}⎤\cdot ⎡\frac{l}{\sqrt{3}\cdot D_{max}}⎤,$$
where $n_{max}$ is the higher limit of the number of waypoints; w and l are width and length, respectively, of the area of interest; and $D_{max}$ is the maximum permitted distance (see Formula (2)).

### 4.2. Optimization of the Positions of Waypoints

The key part of the algorithm in Figure 5 is point 5, which is a stand-alone optimization problem. For every particular deployment of a given number of waypoints, the value of $d_{max}$ can be calculated. The optimization criterion of this problem is expressed by Formula (5). The objective is to deploy the waypoints so that the value of $d_{max}$ is minimal. This consists of finding the positions of n waypoints in a plane, that is, there are 2n independent variables.

(5)$$\mathrm{minimize}\left(d_{max}\right),$$
where $d_{max}$ is the longest existing distance between a point in the area of interest and the nearest waypoint.

The mathematical formulation of the problem is as follows: let $P=\left\{P_{1},P_{2},\ldots \right\}$ be the set of all points that are inside the polygon forming the area of interest; there are an infinite number of points in this set. Let $W=\left\{W_{1},W_{2},\ldots ,W_{n}\right\}$ be the set of n waypoints; and let $D=\left\{d_{1},d_{2},\ldots \right\}$ be the set of real numbers that represent distances between each point $P_{i}\in P$ and the nearest waypoint $W_{\left(i\right)}\in W$, according to Formula (6).

(6)$$d_{i}=\left|P_{i}-W_{\left(i\right)}\right|\;\mathrm{for}\;\mathrm{all}\;P_{i}\in P,$$
where $d_{i}$ is the distance between points $P_{i}$ and $W_{\left(i\right)}$, $P_{i}\in P$ is a point lying inside the area of interest, and $W_{\left(i\right)}\in W$ is the nearest waypoint to point $P_{i}$.

The distance $d_{max}\in D$ is the largest value in set D; see Formula (7).

(7)$$d_{max}=\max D=\max\left(d_{1},d_{2},\ldots \right)$$

#### 4.2.1 Calculation of the Longest Distance

This section deals with the principles of calculating the distance $d_{max}$ formulated above. The value of $d_{max}$ is a function of waypoints $W=\left\{W_{1},W_{2},\ldots ,W_{n}\right\}$, where n≥1 and the polygon of the area of interest $A=\left\{A_{1},A_{2},\ldots ,A_{m}\right\}$. This polygon is represented by the number of m points, where m≥3, thus it is a function of m+n points in a plane (which are 2n+2m variables); see Formula (8).

(8)$$d_{max}=f\left(W,A\right)=f\left(W_{1x},W_{1y},\ldots ,W_{nx},W_{ny},A_{1x,}A_{1y},\ldots ,A_{mx},A_{my}\right)$$

The value of $d_{max}$ is defined as the longest existing distance between a point $P_{i}\in P$ inside polygon A and its nearest waypoint $W_{\left(i\right)}\in W$. As there is an infinite number of points in set P, the longest distance cannot be determined by calculating all distances $d_{i}\in D$.

Instead, a subset C⊂P containing a finite number of points has to be identified. These points $C=\left\{C_{1},C_{2},\ldots ,C_{k}\right\}$ are candidates for becoming points from which the value of $d_{max}$ is calculated using Formula (9).

(9)$$d_{max}=\max\left(\left|C_{1}-W_{\left(1\right)}\right|,\left|C_{2}-W_{\left(2\right)}\right|,\ldots ,\left|C_{k}-W_{\left(k\right)}\right|\right)$$

Points in subset C are identified as follows:

(a)Points that are common to three or more regions of a Voronoi diagram created on waypoints W and, at the same time, which lie inside polygon A.(b)Points that are common to two regions of a Voronoi diagram created on waypoints W
and the border of polygon A.(c)Border points $A_{i}\in A$
of the polygon.

The principle is graphically shown in Figure 6 in an example with 10 waypoints. The green lines represent the polygon of the area of interest, the small yellow dots are waypoints, and the red lines border the regions of the Voronoi diagram. The blue, orange, and green circles show the points of subset C; the colours are used to distinguish between the three cases mentioned above. In this example, the farthest point F from the waypoints is marked in red.

The computational complexity of the proposed principle is mainly influenced by the construction of the Voronoi diagram. This can be done using Fortune’s algorithm with complexity $O\left(n\cdot \log n\right)$, where n is the number of waypoints [30]. The result of this algorithm is a list of edges forming Voronoi regions.

Once the Voronoi diagram has been constructed, the computational complexity of calculating $d_{max}$ for the three cases mentioned above is as follows:

(a)$O\left(n\cdot m\right)$
The points of subset C
are the vertices of the edges of the Voronoi diagram that lie inside the polygon. The number of these vertices is linearly dependent on the number of waypoints n
The test to see whether they are inside the polygon can be done with complexity $O\left(m\right)$
where m
is the number of border points of the polygon. One advantageous feature of Fortune’s algorithm is that each edge is connected with the two waypoints lying in the corresponding regions. These waypoints are the nearest for both vertices and the distance to them can be calculated immediately.(b)$O\left(n\cdot m\right)$
The points of subset C
are created as intersections of the edges of the Voronoi diagram and edges of the polygon. The number of edges of the former is linearly dependent on the number of waypoints n
The number of edges of the latter is identical to the number of points m
of the polygon. Again, as in the previous case, the pair of nearest waypoints for each intersection is known because it is connected with the corresponding edge of the Voronoi diagram.(c)$O\left(n\cdot m\right)$
The points of subset C
are created from m
points of the polygon. For each of these points, the nearest waypoint has to be found with complexity $O\left(n\right)$.

As can be seen, the computational complexity for each of the three cases—(a), (b), and (c)—is the same: $O\left(n\cdot m\right)$. Thus, the total computational complexity of calculating the value of $d_{max}$ for n waypoints and the polygon with m points is given in Formula (10).

(10)$$O\left(n\cdot \log n+n\cdot m\right)$$

#### 4.2.2. Optimization Method

For a solution to this problem, an adapted simulation annealing algorithm is used. This algorithm is a generic probabilistic method and has proven to be very successful when used by the authors in similar position optimization problems; see the works of [31,32].

This algorithm is presented in Figure 7 using pseudocode. A solution x represents a particular deployment of n waypoints, that is, it is a vector of 2n independent variables, where each pair corresponds to a position of one waypoint in a plane: $x=\left(x_{1},x_{2},\ldots ,x_{2n}\right)$.

The key phase of the algorithm is the creation (transformation) of a new solution $x^{\prime }=\left(x_{1}^{\prime },x_{2}^{\prime },\ldots ,x_{2n}^{\prime }\right)$ from the current solution x in point 6. This is done by adding random value(s) with normal distribution to one or more variables of vector x, according to Formula (11).

(11)$$x_{i}^{\prime }=\begin{array}{cc} \left\{\;\begin{array}{l} x_{i}+\mathrm{RandN}\left(\mu ,\sigma^{2}\right)\\ x_{i} \end{array}\right. & \begin{array}{l} \mathrm{for}\;\mathrm{selected}\;\mathrm{variables}\;x_{i}\in x\\ \mathrm{otherwise} \end{array} \end{array},\;\mu =0,\;\sigma =\frac{\left(T-T_{min}\right)\cdot R_{max}}{T_{max}-T_{min}},$$
where RandN is a random number generator with a normal distribution $N\left(\mu ,\sigma^{2}\right),$
T is the current temperature in interval $\langle T_{min},T_{max}\rangle$, and $R_{max}$ is the maximum range in which variable $x_{i}$ should change—this value is connected with the size (width or length) of the area of interest $R_{max}=size/3$.

The decision on selecting which variables to change in Formula (11) depends on the variant of the algorithm. Usually, one of these variants is used:

Select all variables from x
Select variables from x
with probability pSelect just one variable randomly from x. This variant is used for this optimization problem as it was empirically proven to be the variant with the most successful results.

When the new solution $x^{\prime }$ is created, this solution replaces the original solution x with probability $p\left(x\to x^{\prime }\right)$, calculated according to the Metropolis criterion in Formula (12). The Metropolis criterion states that if the new solution is better than the original, the probability of its replacement is 1; otherwise, this probability depends on the difference in their qualities and the current temperature.

(12)$$p\left(x\to x^{\prime }\right)=\begin{array}{cc} \left\{\;\begin{array}{l} 1\\ e^{-\frac{d_{max}^{\prime }-d_{max}}{T}} \end{array}\right. & \begin{array}{l} \mathrm{for}\;d_{max}^{\prime }\leq d_{max}\\ \mathrm{otherwise} \end{array} \end{array},$$
where $d_{max}^{\prime }$ is the longest distance for the new solution $x^{\prime }$ and $d_{max}$ is the longest distance for the original solution x.

#### 4.2.3. Improvement of the Optimization Method

The optimization method introduced in Section 4.2.2 can be further improved for this particular problem. When evaluating the quality of a solution, that is, calculating the value of $d_{max}$ for the given variables of vector x, the position of the furthest point F in the polygon of the area of interest in relation to the waypoints is determined as a by-product (see the red circle in Figure 6 as an example).

Seeing as the aim in resolving the problem is to minimize the distance $d_{max}$ (see Formula (5)), it is likely that a better solution can be created by moving the waypoints (see Formula (11)) that are closer to point F than the others, especially for the later phases of the optimization, when the current temperature is heading towards the minimal temperature $T_{min}$ and changes are smaller.

The algorithm was modified to use this feature as follows: the probability of each variable being selected is calculated according to Formula (13); then, one variable is selected for transformation, not randomly, but based on these probabilities using the roulette wheel principle.

(13)$$p\left(x_{2i-1}\right)=p\left(x_{2i}\right)=\frac{{\left(d_{W_{i}}\right)}^{-1}}{2\cdot \sum_{i}{\left(d_{W_{i}}\right)}^{-1}}\;\mathrm{for}\;\mathrm{each}\;W_{i}\in W,$$
where $p\left(x_{j}\right)$ is the probability of variable $x_{j}\in x$ being selected, j=1,2,…,2n; and $d_{W_{i}}$ is the distance between waypoint $W_{i}\in W$ and point F.

## 5. Experiments and Results

This section presents experiments and results of the proposed algorithm on two sets of benchmark problems.

### 5.1. Benchmark Problems for Verification

The first set of benchmark problems was designed to verify the quality of the proposed probabilistic algorithm. Polygons forming the area of interest for individual problems are created by combining a number of adjacent hexagons; the problems vary from one another according to the number of hexagons combined. The number of waypoints corresponds to the number of hexagons.

The reason for applying these benchmark problems is that it is easy to determine the optimal solution for each, which can then be compared with the solution found by the optimization method. The optimal solution is the solution where waypoints are located at the centres of hexagons. An example of a problem with 17 waypoints (optimally deployed) is shown in Figure 8. The green line demarcates the resulting polygon, while the red lines represent a Voronoi diagram.

Table 1 shows the basic parameters of the benchmark problems. Degree represents the number of hexagons in diagonals of the polygon (see Figure 8).

The proposed optimization algorithms (original and modified versions—see Section 4.2.2 and Section 4.2.3, respectively) were executed 500 times on each benchmark problem. The parameters of the algorithm (see Figure 6) were set as follows: $T_{max}=100$, $T_{min}=10^{-6}$, α=0.95, k=2000, r=200. The experiments were executed on a computer with the following parameters: Intel Core i7-7700 CPU @ 3600 GHz, 32 GB RAM.

Table 2 shows the results achieved by the original simulated annealing algorithm (SA-original) and its modified version (SA-modified). The table records the best solution found, mean and standard deviation for each benchmark problem, as well as the average runtime of a single execution of the algorithm. As can be seen, the modified version is comparable to the original version for simpler problems (v01, v02, v03); however, although the original algorithm is almost twice as fast for v05 and v06 problems, the improvement in solutions of the modified version is clear. The slower runtime of the modified version for more complex problems is caused by calculating the probabilities for selecting variables with computational complexity $O\left(n\right)$ in the solution transformation process (see point 6 in Figure 6), whereas the same process is done with complexity $O\left(1\right)$ in the original version.

Table 3 compares the results. The optimal solution for each benchmark problem is 100 (distance from the centre of any hexagon to its circumscribed circle). The best solutions found by the original and modified algorithms are compared with the optimal solution in the form of percent error. The last column of Table 3 shows the improvement of the modified algorithm compared with the original version. The average percent error for the original and modified algorithms compared with the optimal solution over all six benchmark problems is 0.496% and 0.023%, respectively. The modified version outperforms the original algorithm by 0.473% on average; the results achieved are very close to the optimal solution even for a problem with 71 waypoints, that is, for 142 independent variables.

### 5.2. Benchmark Problems of Typical Reconnaissance Scenarios

The second set of benchmark problems enables a comparison between the new approach to waypoint deployment according to the algorithm in Figure 5 and the original approach in Figure 4. The most important criterion is the reduction in the number of waypoints, the effect of which is to shorten the overall time for a reconnaissance operation.

Table 4 shows the parameters of the benchmark problems, created based on typical reconnaissance situations in the real environment. The table records the basic parameters of the polygon, along with the number of points forming the area of interest and the number of UAVs in the fleet available for the reconnaissance operation. The last column of Table 4 details the maximum permitted distance given by the technical parameters of the particular UAVs and tactical requirements according to Formula (2).

Table 5 shows the optimization of the number of waypoints deployed for the benchmark problems using the algorithm presented Figure 5 compared with the number of waypoints determined using the original algorithm mentioned in Figure 4. The values of the longest distance ($d_{max}$), that is, the solutions, do not exceed the maximum permitted limit ($D_{max}$) in any of the benchmark problems. The last column of Table 5 records the number of waypoints saved by as a result of optimization. On average, the reduction in the number of waypoints is 38.9%.

The computational times of the waypoints’ deployment optimization are also recorded in Table 5. The entire process of planning the reconnaissance operation will be extended by the extra time this process takes, as the computational times of the original waypoints deployment are negligible.

Table 6 compares reconnaissance operations based on the benchmark problems when planned using the original model and the new model in which the number of waypoints and their deployment has been optimized. The optimization criterion was the duration of the operation, that is, the time taken to conduct the whole operation. Table 6 also records the total distance covered by all UASs; however, distance was not the optimization criterion. The last column sets out the reduction in operational durations; the new model managed to shorten reconnaissance operations by 10.3% on average. Also, the total distance decreased on average by 11.2%, the effect of which is lower fuel consumption and the possibility to explore more extensive areas.

To illustrate these findings, Figure 9 shows the benchmark problem r04. Figure 9a describes the result of the original version (82 waypoints). Figure 9b, on the other hand, is the result of the new version (42 waypoints). In Figure 9c, the scenario is shown in the planning software TDSS [1,33], including the visualized terrain and environment.

To prove the effectiveness of the proposed approach for the reconnaissance operations, the results were compared to the lawnmower pattern, which is a common approach when planning UAV trajectories in problems such as reconnaissance or search for a target (Cabreira et al., 2019). The advantage of the lawnmower pattern is that the distance between adjacent flights of UAVs (in rows or columns) could be set to $2\cdot D_{max}$; this is because the trajectory pattern is known in advance (the UAVs do not change their heading in a single column or row).

Table 7 compares the reconnaissance operations planned using the new model and the lawnmower pattern. The total distance covered by all UASs is also recorded, but again, as in the previous case, distance was not the optimization criterion. The difference in the last column of Table 7 shows that the new model outperforms the lawnmower pattern. The difference varies for individual benchmark problems from 7% to almost 60%. This is connected with the quality with which it was possible to divide the area of interest into portions for individual UAVs.

The strength of the new approach proposed in this article is in the balanced and even division of the area of interest into portions for individual UAVs. This even division is almost never possible in the case of the lawnmower pattern, as the whole rows or columns must be visited successively in a straight line. This is illustrated in Figure 10 on the benchmark problem r04, where four UAVs are available. The values $C_{A}$, $C_{B}$, $C_{C}$, $C_{D}$ show the time the corresponding UAV needs to finish its route. The solution is given by the maximum value ($C=\max\left(C_{A},\;C_{B},\;C_{C},\;C_{D},\;\right)$). As can be seen, individual routes are nicely balanced in the case of the new model, as opposed to the lawnmower pattern.

The problem of the balanced division of the area of interest is even more apparent in the case of more complex and structured areas of interest and a larger number of UAVs used (e.g., the polygon bounding the area of interest of benchmark problem r06 has 25 points and 5 UAVs are available; the reconnaissance operation is then almost 60% longer when using the lawnmower pattern).

### 5.3. Simulation Experiment

The experiment based on benchmark problem r01 (see Table 4) was conducted using the Steel Beast Pro simulation system (https://www.esimgames.com). The reason for the experiment was to use simulation in order to prove the validity of the proposed model under various conditions. As UAVs, Lockheed Martin Quad Indago drones were used.

Figure 11 shows routes planned via the original (Figure 11a) and new version (Figure 11b). The model estimates that the operation would be completed, that is, at the moment when both drones had returned to their initial positions, in 6:55 min in the case of the original version (27 waypoints), and in 6:37 min in the case of the new version (18 waypoints). The length of the routes of UAV A is 2.54 km (original version) and 2.26 km (new version), and for UAV B, it is 2.46 km (original version) and 2.43 km (new version); the total distance travelled by both UAVs is 5.00 km (original version) and 4.69 km (new version), respectively.

Table 8 shows the results of the simulations. Overall, three separate experiments (e01, e02, e03) were conducted under various meteorological conditions. In each experiment, five simulations (s01, s02, s03, s04, s05) were carried out, and the average duration of the operation was calculated. In two cases, the simulation was not successful as a result of drone failure. The experiments show the influence of wind speed and precipitation on the flight duration. The wind extends the duration of the operation (compare e01 and e02); furthermore, the rain of medium intensity has a noticeable effect on the duration (e03 takes longer than e02 even if the wind speed is smaller).

Experiment e01 was carried out under ideal weather conditions (sunny, no wind), and thus can be compared to the results estimated by the model (r01)—see Table 9. The time estimated by the model was very close to the average duration of the simulations (difference of about 3%). Moreover, the simulation confirmed the improvement in shortening the duration of the operation when using the new model (about 4% in this case).

## 6. Conclusions

In this paper, the problem of optimizing the number and positions of waypoints in a cooperative aerial reconnaissance model using a number of unmanned aerial systems was examined. As a solution, a modified simulated annealing algorithm was proposed. The improvement of the new algorithm, compared with the principle used in the original CAR model, was verified in a series of experiments.

The experiments in the first set were designed to evaluate the proposed probabilistic algorithm. In these experiments, the optimal deployments of waypoints are already known and thus can be compared with the results achieved by the algorithm. The average difference between the optimal solution and the solution found by the simulated annealing was only 0.023%; in the most complex problem, with 71 waypoints (142 independent variables), the difference from the optimal solution was 0.114%.

The second set of experiments was based on typical reconnaissance scenarios, and their purpose was to evaluate the increase in effectiveness of the reconnaissance operation. The new model managed to shorten the time needed to conduct operations by an average of 10%. Also, the comparison of the results with the lawnmower pattern proved the effectiveness of the proposed method. Furthermore, one of these experiments was performed in the SteelBeast simulation system in order to verify the model under various meteorological conditions. The simulations confirmed the results estimated by the model.

In the CAR model, the collision avoidance of cooperating UAVs is not taken into consideration. It is supposed that each UAV has its own internal collision avoidance system or other precautions are taken (e.g., different height of flight for each vehicle). Moreover, trajectories of individual vehicles should not cross if planned optimally.

The future work of the authors will focus on the problem of persistent surveillance by a number of unmanned aerial vehicles. In this task, similar principles of the optimization of waypoints can be used. Also, the possibility of extending the model to avoid the potential collisions will be investigated. The model could integrate some principles of the traffic collision avoidance systems (TCAS), for example, see the work of [34]. Furthermore, the experiments will be conducted to verify the performance of the model when planning the reconnaissance of a very uneven terrain or in an urban area, where some parts of the area of interest may be occluded.

## Figures and Tables

**Figure 1 sensors-19-02823-f001:**
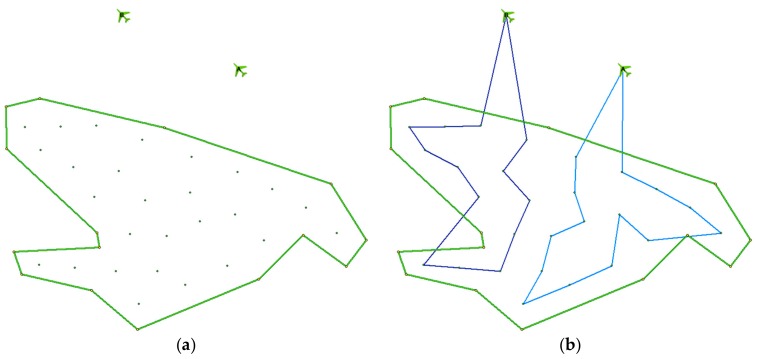
Example situation: (**a**) area of interest covered by waypoints; (**b**) trajectories for individual unmanned aerial systems (UASs).

**Figure 2 sensors-19-02823-f002:**
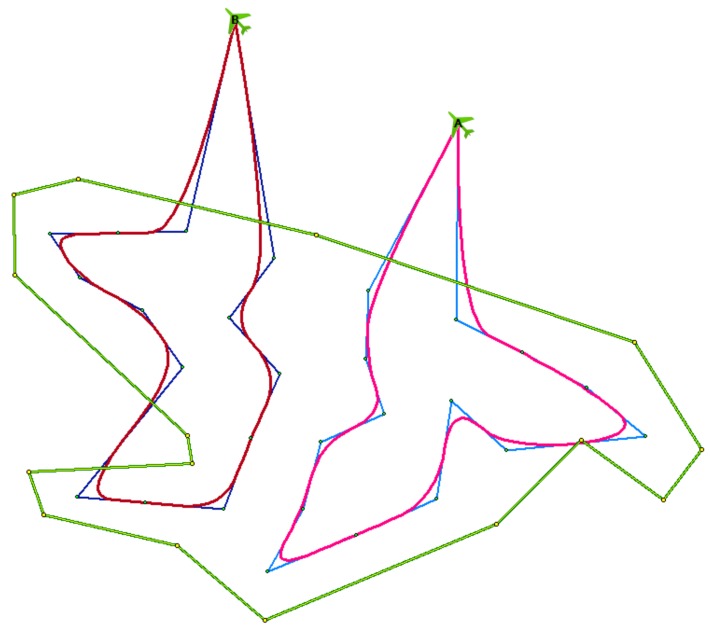
Improvement in the trajectories.

**Figure 3 sensors-19-02823-f003:**
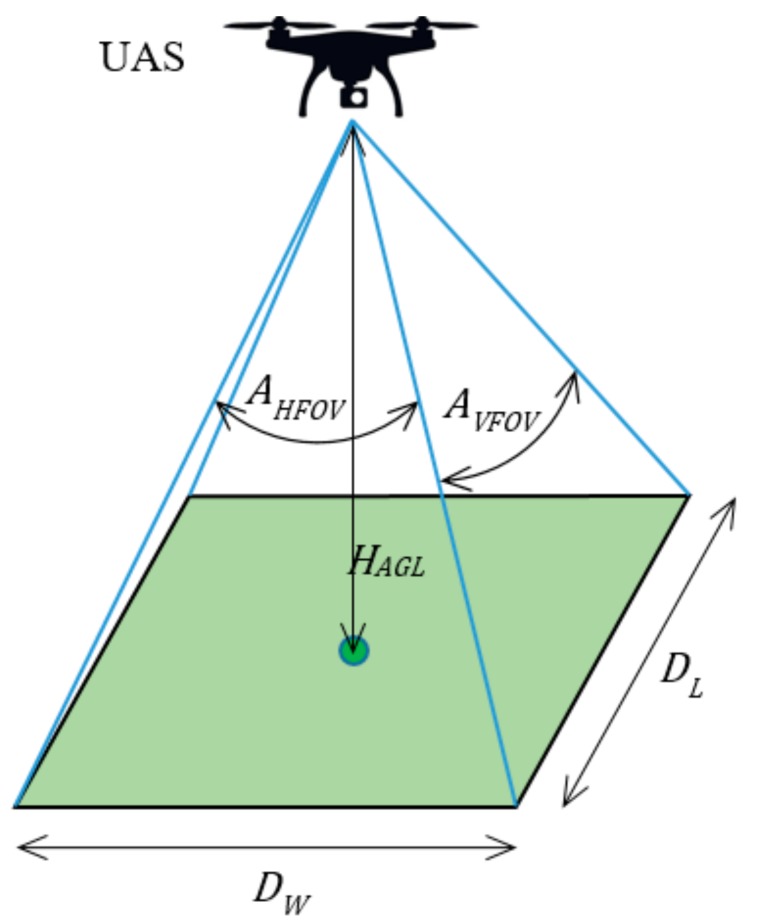
Parameters for deployment of waypoints.

**Figure 4 sensors-19-02823-f004:**
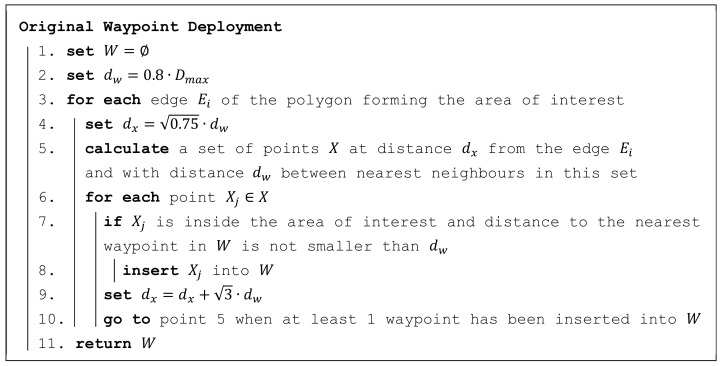
Original waypoint deployment algorithm.

**Figure 5 sensors-19-02823-f005:**
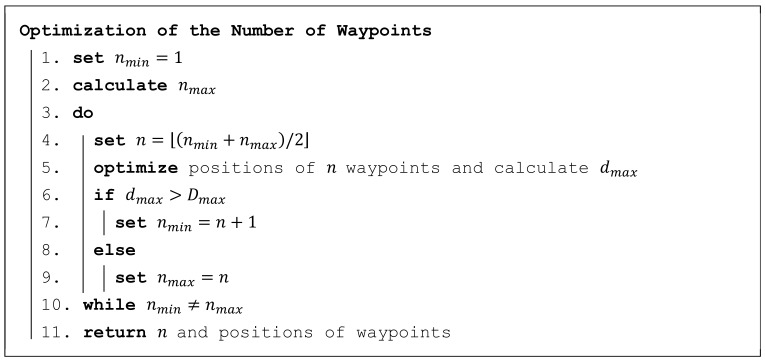
Algorithm for optimization of the number of waypoints.

**Figure 6 sensors-19-02823-f006:**
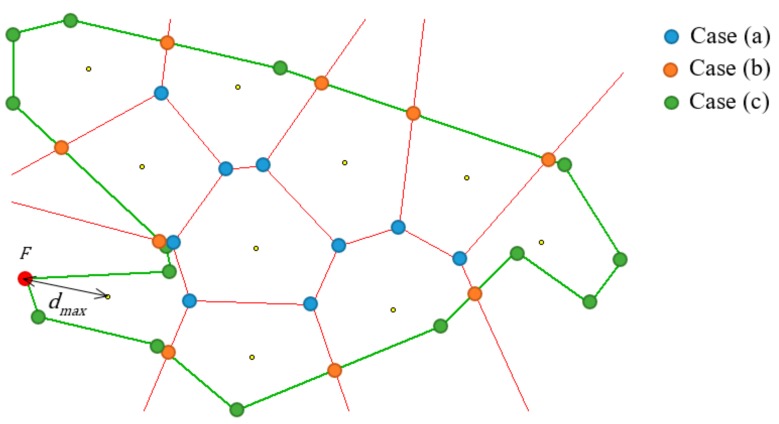
Principle of calculating the longest distance.

**Figure 7 sensors-19-02823-f007:**
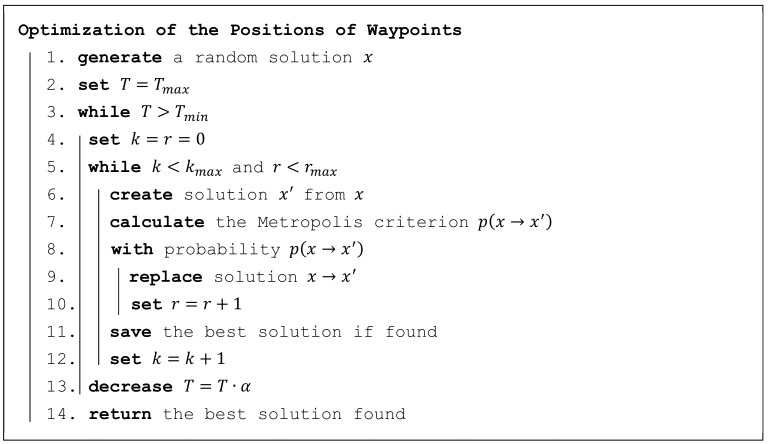
Simulated annealing for the problem of optimizing waypoint positions.

**Figure 8 sensors-19-02823-f008:**
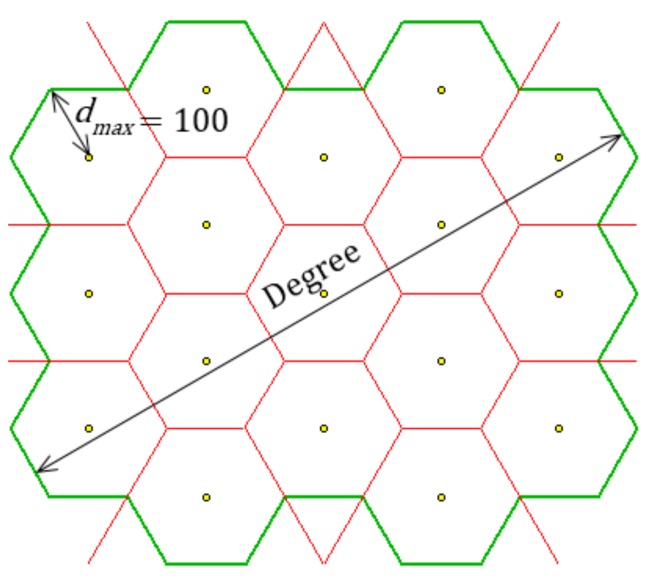
Example of a benchmark problem (v03).

**Figure 9 sensors-19-02823-f009:**
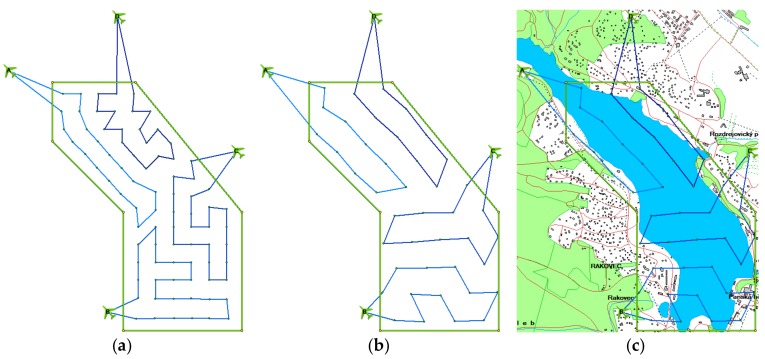
Benchmark problem r04: (**a**) original model; (**b**) new model; (**c**) Tactical Decision Support System (TDSS).

**Figure 10 sensors-19-02823-f010:**
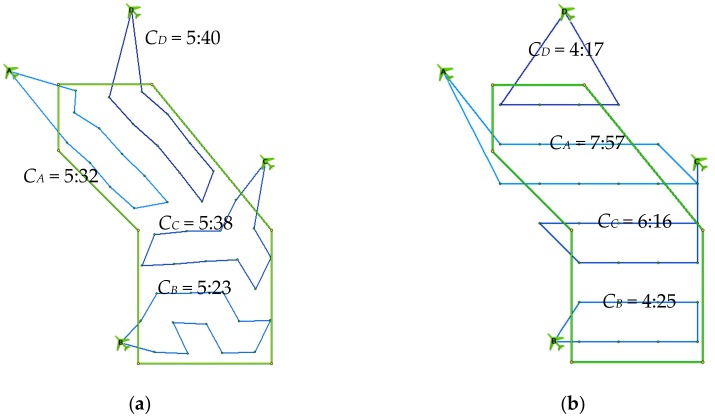
Comparison of the new model with the lawnmower pattern for benchmark problem r04: (**a**) new model; (**b**) lawnmower pattern.

**Figure 11 sensors-19-02823-f011:**
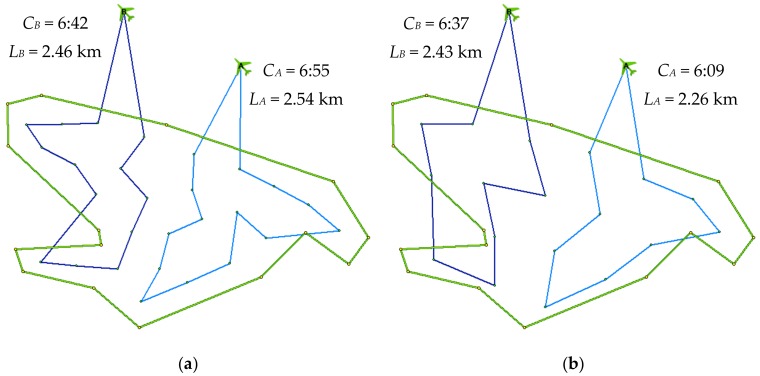
Experiment for simulation (r01): (**a**) original model; (**b**) new model.

**Table 1 sensors-19-02823-t001:** Set of benchmark problems for verification.

Benchmark Problem	Degree	Number of Waypoints	Number of Variables per Solution Vector
v01	1	1	2
v02	3	7	14
v03	5	17	34
v04	7	31	62
v05	9	49	98
v06	11	71	142

**Table 2 sensors-19-02823-t002:** Results for benchmark problems for verification.

Problem	SA-Original				SA-Modified			
	Best	Mean	Stdev	Runtime	Best	Mean	Stdev	Runtime
v01	100.000	100.000	0.000	0.5 sec	100.000	100.000	0.000	0.5 s
v02	100.000	100.004	0.025	5 sec	100.000	100.005	0.033	5 s
v03	100.001	100.064	0.098	12 sec	100.001	100.010	0.022	14 s
v04	100.005	100.884	0.854	25 sec	100.003	100.090	0.230	34 s
v05	100.555	104.904	2.868	38 sec	100.018	100.668	1.594	74 s
v06	102.417	111.278	3.497	53 sec	100.114	103.318	2.377	98 s

**Table 3 sensors-19-02823-t003:** Comparison of results for benchmark problems for verification.

Problem	Optimal Solution	SA-Original		SA-Modified		Algorithm Improvement
		Best	Error	Best	Error	
v01	100	100.000	0.000%	100.000	0.000%	0.000%
v02	100	100.000	0.000%	100.000	0.000%	0.000%
v03	100	100.001	0.001%	100.001	0.001%	0.000%
v04	100	100.005	0.005%	100.003	0.003%	0.002%
v05	100	100.555	0.555%	100.018	0.018%	0.537%
v06	100	102.417	2.417%	100.114	0.114%	2.300%

**Table 4 sensors-19-02823-t004:** Set of benchmark problems for reconnaissance. UAVs—unmanned aerial vehicles.

**Problem**	**Area of Interest**				**Number of UAVs**	***D_max_***
	**Area**	**Width**	**Length**	**Points**		
r01	0.517 km^2^	1.21 km	0.78 km	15	2	120 m
r02	0.679 km^2^	0.95 km	1.14 km	5	2	100 m
r03	2.640 km^2^	2.20 km	1.20 km	4	1	150 m
r04	2.085 km^2^	1.60 km	2.10 km	7	4	150 m
r05	1.675 km^2^	2.15 km	1.25 km	9	4	100 m
r06	2.757 km^2^	3.75 km	3.04 km	25	5	120 m

**Table 5 sensors-19-02823-t005:** Optimization of the number of waypoints for reconnaissance benchmark problems.

**Problem**	**Original Deployment**		**Optimized Deployment**			**Reduction in Waypoints**
	**Waypoints**	$d_{max}$	**Waypoints**	$d_{max}$	**Runtime**	
r01	27	119.712	18	118.295	12 sec	9 (33.3%)
r02	59	99.162	31	99.058	16 sec	28 (47.5%)
r03	66	141.421	52	149.668	29 sec	14 (21.2%)
r04	82	149.771	42	148.767	25 sec	40 (48.8%)
r05	124	99.556	75	99.552	50 sec	49 (39.5%)
r06	166	119.832	94	119.632	103 sec	72 (43.4%)

**Table 6 sensors-19-02823-t006:** Reconnaissance operations planned using the original and new models.

**Problem**	**Original Deployment**		**Optimized Deployment**		**Reduction in Waypoints**
	**Duration**	**Distance**	**Duration**	**Distance**	
r01	6:55	5.00 km	6:37	4.69 km	4.3%
r02	12:57	7.77 km	10:39	6.31 km	17.8%
r03	22:42	13.62 km	21:32	12.93 km	5.1%
r04	6:35	15.65 km	5:40	13.34 km	13.9%
r05	6:20	15.17 km	5:39	13.48 km	10.8%
r06	10:26	31.28 km	9:25	27.84 km	9.7%

**Table 7 sensors-19-02823-t007:** Reconnaissance operations planned using the new model and the lawnmower pattern.

**Problem**	**New Model**		**Lawnmower Pattern**		**Difference**
	**Duration**	**Distance**	**Duration**	**Distance**	
r01	6:37	4.69 km	7:05	4.85 km	7.1%
r02	10:39	6.31 km	15:33	7:59 km	46.0%
r03	21:32	12.93 km	22:45	13.65 km	5.7%
r04	5:40	13.34 km	7:57	13.75 km	40.3%
r05	5:39	13.48 km	6:20	13.57 km	12.1%
r06	9:25	27.84 km	14:52	33.90 km	57.9%

**Table 8 sensors-19-02823-t008:** Results of the simulation experiment.

**Experiment**	**e01** **Sun** **No Wind**		**e02** **Overcast** **Fresh Breeze (30 km/h)**		**e03** **Medium Rain** **Moderate Breeze (20 km/h)**	
**Simulation**	**Original**	**New**	**Original**	**New**	**Original**	**New**
s01	7:25	7:11	9:15	8:25	10:04	9:22
s02	7:11	6:54	8:05	8:14	9:48	8:47
s03	6:54	6:12	8:55	8:54	Failure	9:38
s04	7:14	6:55	8:45	9:01	9:34	8:26
s05	6:47	7:02	Failure	8:27	9:47	9:11
Average	7:06	6:50	8:45	8:36	9:48	9:04

**Table 9 sensors-19-02823-t009:** Comparison of estimated and simulation results.

**Experiment**	**Original Model**	**New Model**	**Improvement**
e01	7:06	6:50	3.61%
r01	6:55	6:37	4.34%
Difference	2.63%	3.36%	0.72%

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
