# Peer review of "Trajectory Optimization in a Cooperative Aerial Reconnaissance Model"

_sensors, 2019, doi:10.3390/s19122823_

Round 1
Reviewer 1 Report
The organization of this paper must be re-organized. I can hardly understand what is the authors' work in this paper.
Author Response
Dear reviewer, thank you for you comments. The response is in the file colored in red.

Reviewer 2 Report
This article examines a model for planning aerial reconnaissance by a fleet of 14 unmanned aerial vehicles that are mutually cooperating to increase the effectiveness of the task. When multiple UAVs conduct reconnaissance in a certain airspace, the collision avoidance should be taken into consideration.The author should take the paper "A causal encounter model of traffic collision avoidance system operations for safety assessment and advisory optimization in high density airspace" into consideration.
Author Response
Dear reviewer, thank you very much for you comments. The response is in the file colored in red.

Reviewer 3 Report
The paper describes a method for defining optimal search paths for a fleet of unmanned aircraft. The method places a set of waypoints so that the path defined by those waypoints ensures coverage of a desired set of search locations. The cost function can be defined by the user; in this case, minimum path length is used. The authors compare an improved approach (compared with an earlier method published by the authors) of defining the waypoint locations; here, improved means that the total path length (and by extension, time required to complete the search) is shorter.
The paper is well written and easy to follow.
General comments
Optimal search has been a very active area of research in the mobile robot community for decades. Often grid based methods are used (see for example Bourgault et al. "Optimal Search for a Lost Target in a Bayesian World, https://link.springer.com/chapter/10.1007/10991459_21 and Hollinger et al. "Efficient multi-robot search for a moving target," https://journals.sagepub.com/doi/abs/10.1177/0278364908099853?journalCode=ijra). A significant number of citations are to the authors' own prior work, and it is not clear that the authors are aware of the existing literature in optimal search.
It is thus not entirely clear that there is a novel contribution here... the authors have improved on their earlier method, but what is the motivation for using a waypoint-based to define the search pattern? If the author's approach could run in useful time on a computer that can be carried on a small UAS then there is a contribution, but the authors' results have been run on a desktop-scale computer (Intel Core i7-7700 CPU @ 3600 GHz, 32 GB RAM, from page 10, line 321). At the least, the authors should compare their search results with a lawnmower pattern and a spiral search pattern.
Page 2, line 72: A star is normally written A*
Page 2, line 78: "Meanwhile" should not be capitalized
Section 5.2
The authors should mention solution times for the original and new approaches in tables 5 and 6. This is a key parameter... a solution that takes hours (or days) to compute may not be helpful (for example, a sub-optimal solution that is computed very quickly can be deployed and completed in shorted overall time than an optimal but slow-to-compute solution) or it may be obsolete because of changes in weather or some other parameter.
Section 5.3
The distances and durations in Table 6 suggest an average speed of roughly 42 km/h (roughly 23 knots, or 11.7 m/s).
Table 7 defines Fresh Breeze and Moderate Breeze as 30 m/s (58 knots) and 20 m/s (38.9 knots), respectively. Is this a typo? if not, there are serious questions that must be addressed. Both quoted wind speeds are faster than the apparent air speed of the quad rotor (from Table 6). How can the drone complete the search task when it cannot make headway? Note that according to the Beaufort Wind Scale, a fresh breeze is 17-21 knots (8 - 10.7 m/s) and a moderate breeze is 11-16 knots (5.5 - 7.9 m/s). Even at these wind speeds, the change in mission duration between no-wind and wind does not seem to match: the upper limit of fresh breeze is close to the flight speed of the aircraft, so it's up-wind travel speed will be very low. Even if the wind is a cross-wind, the resulting ground speed will be significantly lower than the airspeed. Note that if winds were 3 m/s and 2 m/s, the results may make sense. It would be very helpful to see the change in search pattern for a given wind condition.
Ref. 16 is incorrect... it should be
Geiger, B. Unmanned Aerial Vehicle Trajectory Planning with Direct Methods, PhD thesis, Pennsylvania State University, University Park, PA, 2009.
Author Response

(The authors gave the same response as above.)

Round 2
Reviewer 1 Report
The authors have made a good response to reviewers' questions. Please conseder enrish the context of this paper by citing the following reference:
[R] S. Fu, J. Wu, H. Wen, Y. Cai, and B. Wu, “Software defined wireline-wireless cross-networks: Framework, challenges, and prospects,” IEEE Communications Magazine, vol. 56, no. 8, pp. 145–151, 2018.
Author Response
Dear reviewer, thank you for your review and valuable comments!
Reviewer 3 Report
Review of v2 of Stodola et al., "Trajectory Optimization in a Cooperative Aerial Reconnaissance Model"
The paper describes a method for defining optimal search paths for a fleet of unmanned aircraft. The method places a set of waypoints so that the path defined by those waypoints ensures coverage of a desired set of search locations. The cost function can be defined by the user; in this case, minimum path length is used. The authors compare an improved approach (compared with an earlier method published by the authors) of defining the waypoint locations; here, improved means that the total path length (and by extension, time required to complete the search) is shorter.
The authors have addressed most of my concerns. Seeing runtimes is very helpful, and it is helpful to see the improvement over a lawnmower pattern.
The authors have corrected a units typo in wind speeds. I am still puzzled by the results in Section 5.3, however. Without knowing the flight speeds of the drones in the simulation (airspeed and ground speed) as well as the energy consumed it is impossible to tell if the results are correct and/or feasible for deployment. As a demonstrative example: I suspect that a 30 km/h wind is at least half the normal cruising airspeed of the drones used in the simulation. Assuming a total distance of 20 km on an out-and-return mission, and assume there is a 30 km/h wind (tailwind on the way out). At an airspeed of 30 km/h, the drone will never return to base. At an airspeed of 60 km/h, the outbound ground speed is 90 km/h and the return ground speed is 30 km/h... so the outbound time is 6 minutes 40 seconds and the return time is 20 minutes. Does the drone feasibly have an endurance of 26 minutes 40 seconds at an airspeed of 60 km/h? I am aware that one would change the cruising airspeed to mitigate the effect of winds (compute the optimal airspeed to minimize the energy consumed per distance travelled).
I see in the author's response that the distance reported is the total distance covered by all vehicles... reporting the longest route and the shortest route would help address my question about range and endurance in wind (if the longest distance flown is only a couple of kilometers, then there is no concern). I would still love to see a side-by-side comparison of flight paths in no wind and with 30 km/h wind (e.g. repeat Figure 9, but with the no-wind paths and the 30 km/h wind paths).
How is it possible that the 20 km/h results all show a longer time than the 30 km/h results? Since all the paths are closed, increasing wind will necessarily lead to longer flight times... even if flight speeds are being optimized for energy consumption in winds. Perhaps the columns were swapped?
I'm still listing this as "accept after major revisions" because I think they will make a big difference in the overall quality of the paper. However, I expect that the authors have the data easily accessible, so I don't think it will be a large burden to implement the suggestions.
Author Response
Dear reviewer, thank you for your review and valuable comments!
The response to them is in the file (all changes are commented).

Round 3
Reviewer 3 Report
The authors have sufficiently addressed my concerns within the limits of their simulation environment.
I suggest that the authors develop/obtain a simulation environment that gives them access to vehicle states and energy usage... Gazebo is quite popular among the robotics community, FlightGear can be very useful for "aircraft oriented" simulations.
Author Response
Thank you very much for your hard work and useful recommendations. We are convinced that our atricle has been improved a lot.